# Retrieving Multimodal Information for Augmented Generation: A Survey

**Ruochen Zhao**[1]    **Hailin Chen**[1]    **Weishi Wang**[1]    **Fangkai Jiao**[1]
**Xuan Long Do**[1*]    **Chengwei Qin**[1]    **Bosheng Ding**[1]    **Xiaobao Guo**[1]
**Minzhi Li** [2]    **Xingxuan Li** [1]    **Shafiq Joty**[1,3†]

[1]Nanyang Technological University, Singapore
[2]National University of Singapore, Singapore
[3]Salesforce Research

{ruochen002, hailin001, xuanlong001, weishi001, fangkai002, chengwei003, bosheng001}@e.ntu.edu.sg

{xiaobao001, xingxuan001}@e.ntu.edu.sg, li.minzhi@u.nus.edu, srjoty@ntu.edu.sg

## Abstract

As Large Language Models (LLMs) become popular, there emerged an important trend of using multimodality to augment the LLMs' generation ability, which enables LLMs to better interact with the world. However, there lacks a unified perception of at which stage and how to incorporate different modalities. In this survey, we review methods that assist and augment generative models by retrieving multimodal knowledge, whose formats range from images, codes, tables, graphs, to audio. Such methods offer a promising solution to important concerns such as factuality, reasoning, interpretability, and robustness. By providing an in-depth review, this survey is expected to provide scholars with a deeper understanding of the methods' applications and encourage them to adapt existing techniques to the fast-growing field of LLMs.

## 1   Introduction

Generative Artificial Intelligence (GAI) has demonstrated impressive performances in tasks such as text generation (Ouyang et al., 2022; Chowdhery et al., 2022; Brown et al., 2020) and text-to-image generation (Ramesh et al., 2021a; Poole et al., 2022). The recent advancements in Multimodal Large Language Models (MLLMs) (Driess et al., 2023; OpenAI, 2023; Huang et al., 2023b) have further improved the models' capabilities to handle multi-format information, opening up possibilities for developing general-purpose learners.

Nevertheless, generative models are not exempt from inherent limitations, including the tendency for generating hallucinations (Ye and Durrett, 2022), struggling with arithmetic tasks (Patel et al., 2021), and lacking interpretability . Consequently, a promising solution for enhancing their capabilities lies in enabling them to interact with the external world and acquire knowledge in diverse formats and modalities, thereby improving the factuality and rationality of the generated content. Recently, there have been emerging studies focusing on retrieval-augmented approaches (Mialon et al., 2023), which aim to provide information that is more grounded and factually dependent. Among them, most (Nakano et al., 2021; Guu et al., 2020b) retrieves textual information, which matches the data format used during pre-training and offers a natural medium for interaction. However, there is more world knowledge stored in different structures and modalities, such as images and videos, which is often inaccessible, unavailable, or not describable in traditional textual corpora.

Therefore, there arises an important research intersection that retrieves multimodal knowledge to augment generative models. It offers a promising solution to current challenges such as factuality, reasoning, interpretability, and robustness. As this field is very recent, there lacks a unified understanding of recognizing these methods as a specific group, visualizing their innate connections, connecting their methodologies, and outlining their applications.

Therefore, we survey recent advancements in multimodal retrieval-augmented generation (RAG). Specifically, we discuss current research by grouping them into different modalities, including image, code, structured knowledge, audio, and video. For each modality, we systematically search the ACL Anthology and Google Scholar with relevant keywords and perform manual filtering to determine their relevance to the survey. As a result, we collect 146 papers for detailed analysis. In Appendix A.1, we include search details, statistics, and a trend analysis figure, which shows that multimodal RAG papers have indeed developed very fastly since the emergence of large-scale general-purpose models. Within each modality, we discuss relevant papers by grouping them under different applications. By providing an in-depth survey, we hope to help researchers recognize the importance

---

*Now affiliated with NUS
†Work done while the author is on leave from NTU

of incorporating knowledge in different formats and encourage adaptation and advancements on existing techniques to the fast-growing field of LLMs.

In summary, our contributions are as follows:

- We establish retrieval augmented generation with multi-modality as an important group of methods that emerges with the recent advances in LLMs.
- For common modalities, we provide an in-depth review of research papers by contextualizing their innate connections and shared challenges.
- We provide an informative analysis of future directions, which could contain promising solutions to many current challenges.

## 2 Definitions and Background

To better understand the state and advancements that inspired multimodal retrieval augmentation, we first define and discuss the background of two key concepts: multimodal learning and retrieval-augmented generation (RAG).

### 2.1 Multimodal Learning

Multimodal learning refers to learning a unified representation of data from different modalities. It aims at extracting complementary information to facilitate compositional tasks (Baltrušaitis et al., 2018; Gao et al., 2020). In this survey, we include all modalities whose formats are different from natural language, including image, code, structured knowledge (*e.g.* tables, knowledge graphs), audio, and video.

Multimodal generative models have a wide range of applications, such as text-image generation, creative writing generation, and multilingual translation. For instance, the image recognition task can benefit from analyzing images and videos in conjunction with textual descriptions (Ju et al., 2022; Alayrac et al., 2022a; Jia et al., 2021; Radford et al., 2021b). Conversely, incorporating visual information also aids language understanding and generation (Zhou et al., 2020; Lei et al., 2021; **?**). Moreover, they have the potential to significantly improve machine learning systems across various domains by enabling models to learn from and integrate multiple sources of information (Tsai et al., 2019; Acosta et al., 2022; Nagrani et al., 2021). Additionally, there has been growing interest in developing generative models that can output multiple modalities of data (Ramesh et al., 2021b; Crowson et al., 2022; Lin and Byrne, 2022a; Chen et al., 2022a). However, there remain challenges for multimodal generative models, such as gaining access to a large amount of multimodal data and designing a network that produces semantically meaningful outputs.

### 2.2 Retrieval-Augmented Generation (RAG)

RAG typically consists of two phases: retrieving contextually relevant information, and guiding the generation process using the retrieved knowledge.

Recently, RAG has gained popularity in Natural Language Processing (NLP) due to the rise of general-purpose Large Language Models (LLMs) (Chowdhery et al., 2022; OpenAI, 2023), which have boosted performances in a wide range of NLP tasks. However, there are two primary challenges: Firstly, because generative models rely on the inner knowledge (weights), they result in a high amount of hallucinations (Zhao et al., 2023b). Secondly, due to their large parameter sizes and the high costs of updating, the traditional pretraining and finetuning approaches have become infeasible. As a solution, RAG methods (Gu et al., 2018; Weston et al., 2018; Cai et al., 2019b; Lewis et al., 2020) offer a promising solution for LLMs to effectively interact with the external world.

RAG is applied to a wide range of downstream NLP tasks, including machine translation (Gu et al., 2018; Zhang et al., 2018; Xu et al., 2020; He et al., 2021), dialogue generation (Weston et al., 2018; Wu et al., 2019; Cai et al., 2019a), abstractive summarization (Peng et al., 2019), and knowledge-intensive generation (Lewis et al., 2020; Izacard and Grave, 2021). Among them, most methods focus on retrieving textual information. For example, Guu et al. (2020b); Lewis et al. (2020); Borgeaud et al. (2022); Izacard et al. (2022) jointly train a retrieval system with an encoder or sequence-to-sequence LM, achieving comparable performance to larger LMs that use significantly more parameters. Recent research also proposes combining a retriever with chain-of-thought (CoT) prompting for reasoning to augment language models (He et al., 2022a; Trivedi et al., 2022; Zhao et al., 2023c).

## 3 Multimodal Retrieval-Augmented Generation

For each modality, there are different retrieval and synthesis procedures, targeted tasks, and challenges. Therefore, we discuss relevant methods by grouping them in terms of modality, including image, code, structured knowledge, audio, and video.

## 3.1 Image

Recent advances on pretrained models shed light on general image-text multi-modal models (Ramesh et al., 2021a; Alayrac et al., 2022b; Aghajanyan et al., 2022; Yu et al., 2022; Dou et al., 2022; Li et al., 2023a). However, these models require huge computational resources for pretraining and large amounts of model parameters — as they need to memorize vast world knowledge. More critically, they cannot efficiently deal with new or out-of-domain knowledge. To this end, multiple retrieval-augmented methods have been proposed to better incorporate external knowledge from images and text documents. In general text generation tasks, image retrieval can also improve generation quality by expanding text generation contexts with more "imagination".

**Visual question answering (VQA)** To tackle open-domain VQA, RA-VQA (Lin and Byrne, 2022b) jointly trains the document retriever and answer generation module by approximately marginalizing predictions over retrieved documents. It first uses existing tools of object detection, image captioning, and optical character recognition (OCR) to convert target images to textual data. Then, it performs dense passage retrieval (DPR) (Karpukhin et al., 2020a) to fetch text documents relevant to the target image in the database. Finally, each retrieved document is concatenated with the initial question to generate the final prediction, similar to RAG (Lewis et al., 2020). Besides external documents, PICa (Yang et al., 2022b) and KAT (Gui et al., 2022) also consider LLMs as implicit knowledge bases and extract relevant implicit information from GPT-3. Plug-and-Play (Tiong et al., 2022) retrieves relevant image patches by using GradCAM (Selvaraju et al., 2017) to localize relevant parts based on the initial question. It then performs image captioning on retrieved patches to acquire augmented context. Beyond text-only augmented context, MuRAG (Chen et al., 2022b) retrieves both text and image data and incorporates images as visual tokens. RAMM (Yuan et al., 2023) retrieves similar biomedical images and captions and encodes them through different networks.

**Image captioning** To generate multi-style captions, Zhou and Long (2023) uses a style-aware visual encoder to retrieve image contents before generating captions. Beyond simply encoding visual information, Cho et al. (2022) further uses the multimodal similarity between image-text pairs as

a reward function to train a more fine-grained captioning model. Beyond retrieving image elements, Sarto et al. (2022); Shi et al. (2021); Ramos et al. (2023); Yang et al. (2023b) retrieves relevant captions to the inputs. Zhou et al. (2022a) tackles news image captioning by retrieving visually grounded entities in news articles.

**Visually grounded dialogue** This task (Lee et al., 2021b) requires retrieving visual information to produce relevant dialog responses. Fan et al. (2021) augments generative models with KNN-based Information Fetching (KIF) modules that retrieve images and wiki knowledge. Liang et al. (2021) retrieves a correlated image to the dialog from an image index to ground the response generator. Shen et al. (2021) trains a word-image mapping model to retrieve response visual impressions and then uses both textual and visual information for response generation.

**Text generation** For general text generation tasks, image retrieval can also help expand contexts. Yang et al. (2022a) augments a text model's "imagination" by retrieving existing images and synthesizing newly generated images. As a result, fueling language models with imagination can improve performances in many downstream natural language tasks. Similarly, Zhu et al. (2023) compares "imagination" augmentation with synthetic and retrieved images and argues that machine-generated images could provide better guidance due to better consideration of the contexts. Moreover, Fang and Feng (2022) shows that machine translation can be significantly improved by retrieving visual information at the phrase level, especially when the textual context is limited. Image RAG can also help low-resource tasks such as medical report generation (Wu et al., 2022a) and architectural description generation (Mille et al., 2020).

Beyond retrieving images before generating text, Re-Imagen (Chen et al., 2022c) leverages a multi-modal knowledge base to retrieve image-text pairs to facilitate image generation. RA-CM3 (Yasunaga et al., 2022) can generate mixtures of images and text. It shows that retrieval-augmented image generation performs much better on knowledge-intensive generation tasks and opens up new capabilities such as multimodal in-context learning.

## 3.2 Code

Software developers attempt to search for relevant information to improve their productivity from large amounts of available resources such as expla-

nations for unknown terminologies, reusable code patches, and solutions to common programming bugs (Xia et al., 2017). Inspired by the progress of deep learning in NLP, a general retrieval-augmented generation paradigm has benefited a wide range of code intelligent tasks, including code completion (Lu et al., 2022b), code generation (Zhou et al., 2022b), and automatic program repair (APR) (Nashid et al., 2023). However, these approaches often treat programming languages and natural languages as equivalent sequences of tokens and ignore the rich semantics inherent to source code. To address these limitations, recent research work has focused on improving code generalization performance via multimodal learning, which incorporates additional modalities such as code comments, identifier tags, and abstract syntax trees (AST) into code pretrained models (Wang et al., 2021b; Guo et al., 2022; Li et al., 2022d). To this end, multimodal retrieval-augmented generation approach has demonstrated its feasibility in a variety of code-specific tasks.

**Text-to-Code Generation**    Numerous research studies have investigated the utilization of relevant codes and associated documents to benefit code generation models. A prominent example is RED-CODER (Parvez et al., 2021), which retrieves the top-ranked code snippets or summaries from an existing codebase, and aggregates them with source code sequences to enhance the generation or summarization capabilities. As another such approach, DocPrompting (Zhou et al., 2022b) uses a set of relevant documentation as in-context prompts to generate corresponding code via retrieval. In addition to these lexical modalities, Hayati et al. (2018) proposes a syntax-based code generation approach to reference existing subtree from the AST as templates to direct code generation explicitly.

**Code-to-Text Generation**    Retrieval-based code summarization methods are studied extensively. For example, RACE (Shi et al., 2022) leverages relevant code differences and their associated commit messages to enhance commit message generation. Besides, RACE calculates the semantic similarity between source code differences and retrieved ones to weigh the importance of different input modalities. Rencos (Zhang et al., 2020) retrieves two similar code snippets based on the aspects of syntactic-level similarity and semantic-level similarity of a given query code. These similar contexts are then incorporated into the summarization model during

the decoding phase. This idea is further explored by Liu et al. (2021), where retrieved code-summary pairs are used to augment the original code property graph (Yamaguchi et al., 2014) of source code via local attention mechanisms. To capture the global semantics for better code structural learning, a global structure-aware self-attention mechanism (Zhu et al., 2019) is further employed.

**Code Completion**    Recent advances in retrieval-based code completion tasks (McConnell, 2004) have gained increasing attention. Notably, Hashimoto et al. (2018) adapts the retrieve-and-edit framework to improve the model's performance in code auto-completion tasks. To address practical code completion scenarios, ReACC (Lu et al., 2022b) takes both lexical and semantic information of the unfinished code snippet into account, utilizing a hybrid technique to combine a lexical-based sparse retriever and a semantic-based dense retriever. First, the hybrid retriever searches for a relevant code from the codebase based on the given incomplete code. Then, the unfinished code is concatenated with the retrieval, and an auto-regressive code completion generator will generate the completed code based on them. In order to address project relations, CoCoMIC (Ding et al., 2022) decomposes a code file into four components: *files*, *global variables*, *classes*, and *functions*. It constructs an in-file context graph based on the hierarchical relations among all associated code components, forming a project-level context graph by considering both in-file and cross-file dependencies. Given an incomplete program, CoCoMIC retrieves the most relevant cross-file entities from its project-level context graph and jointly learns the incomplete program and the retrieved cross-file context for code completion.

**Automatic Program Repair (APR)**    Inspired by the nature that a remarkable portion of commits is comprised of existing code commits (Martinez et al., 2014), APR is typically treated as a search problem by traversing the search space of repair ingredients to identify a correct fix (Qi et al., 2014), based on a redundancy assumption (White et al., 2019) that the target fix can often be reconstructed in the search space. Recent studies have shown that mining relevant bug-fix patterns from existing search space (Jiang et al., 2018) and external repair templates from StackOverflow (Liu and Zhong, 2018) can significantly benefit APR

models. Joshi et al. (2022) intuitively ranks a collection of bug-fix pairs based on the similarity of error messages to develop few-shot prompts. They incorporate compiler error messages into a large programming language model Codex (Chen et al., 2021) for multilingual APR. CEDAR (Nashid et al., 2023) further extends this idea to retrieval-based prompts design using relevant code demonstrations, comprising more modalities such as unit test, error type, and error information. Additionally, Jin et al. (2023) leverage a static analyzer Infer to extract error type, error location, and syntax hierarchies (Clement et al., 2021) to prioritize the focal context. Then, they retrieve semantically-similar fixes from an existing bug-fix codebase and concatenate the retrieved fixes and focal context to form the instruction prompts for program repair.

**Reasoning over Codes as Intermediate Steps**
While large language models (LLMs) have recently demonstrated their impressive capability to perform reasoning tasks, they are still prone to logical and arithmetic errors (Gao et al., 2022a; Chen et al., 2022d; Madaan et al., 2022). To mitigate this issue, emerging research papers have focused on using LLMs of code (e.g., Codex (Chen et al., 2021)) to generate the code commands for solving logical and arithmetic tasks and calling external interpreters to execute the commands to obtain the results. Notably, Gao et al. (2022a) proposes to generate Python programs as intermediate reasoning steps and offload the solution step to a Python interpreter. Additionally, Chen et al. (2022d) explore generating chain-of-thought (CoT) (Wei et al., 2022) for not only text but also programming language statements as reasoning steps to solve the problem. During the inference phase, answers are obtained via an external interpreter. Similarly, Lyu et al. (2023) propose Faithful CoT that first translates the natural language query to a symbolic reasoning chain and then solves the reasoning chain by calling external executors to derive the answer. Another example is Ye et al. (2023), which utilizes LLMs to decompose table-based reasoning tasks into subtasks, decouples logic and numerical computations in each step through SQL queries by Codex, and calls SQL interpreters to solve them (a process called "parsing-execution-filling").

LLMs of code are also known as good-structured commonsense reasoners, and even better-structured reasoners than LLMs (Madaan et al., 2022). As a result, prior studies have also investigated the idea of transforming structured commonsense generation tasks into code generation problems and employing LLMs of code as the solvers. One such work is CoCoGen (Madaan et al., 2022) which converts each training sample (consisting of textual input and the output structure) into a Tree class in Python. The LLMs of code then perform few-shot reasoning over the textual input to generate Python code, and the Python code is then converted back to the original structure for evaluation. Besides, the success of LLMs of code such as Codex in synthesizing computer code also makes them suitable for generating formal codes. Motivated by this, Wu et al. (2022b) propose to prove mathematical theorems by adopting Codex to generate formalized theorems from natural language mathematics for the interactive theorem prover Isabelle (Wenzel et al., 2008).

## 3.3 Structured Knowledge

An open challenge in generative models is hallucination, where the model is likely to output false information. Thus, A potential solution is to ground generation with retrieved structured knowledge, such as knowledge graphs, tables, and databases.

**Question Answering (QA)**  A natural setting to use knowledge is QA. To augment *Knowledge Base (KB) QA* by extracting the most relevant knowledge, Hu et al. (2022b) uses dense retrieval and Liu et al. (2022b) uses a cross-encoder ranker. Shu et al. (2022) employs multi-grained retrieval to extract relevant KB context and uses constrained decoding to control the output space. In *table QA*, Nan et al. (2022) proposes a dataset that requires retrieving relevant tables for answer generation. Pan et al. (2021) then proposes a method that uses a transformer-based system to retrieve the most relevant tables and locate the correct cells. Moreover, to improve *Video QA*, Hu et al. (2023) retrieves from Knowledge Graph (KG) encodings stored in the memory. The most prominent RAG usage remains in *open-domain QA*, where multiple datasets are proposed (Lin et al., 2023; Ramnath et al., 2021). For retrieval, Ma et al. (2022) verbalizes the KG and then uses dense passage retrieval. Fan et al. (2019); Gupta et al. (2018) encodes KG information into dense representations. Pramanik et al. (2021); Jin et al. (2022) builds graph embeddings to retrieve question-relevant evidence. Xu et al. (2021); Baek et al. (2023) use semantic similarity and text-matching methods. Synthesis can occur at different stages. At the input stage, Xu

et al. (2021); Baek et al. (2023) feed in the retrieved contexts as additional inputs or prompts to the PLM. (Ma et al., 2022; Fan et al., 2019) adapt the generator to accept the context representations as inputs. At model inference stage, an interesting work is Hu et al. (2022c), which inserts an interaction layer into PLMs to guide an external KG reasoning module.

**General text generation**    External knowledge retrieval can improve general text generation to be more factually grounded. Liu et al. (2022a) presents a memory-augmented approach to condition an autoregressive language model on a knowledge graph (KG). During inference, Tan et al. (2022) selects knowledge entries through dense retrieval and then injects them into the input encoding and output decoding stages in pretrained language models (PLMs). For *domain-specific text generation*, Frisoni et al. (2022); Yang et al. (2021); Li et al. (2019) retrieve medical report chunks or report templates to augment input prompts. Then, they use self-devised decoders or graph transformers to generate grounded reports. To improve interpretability, RAG could be used to select facts as interpretable reasoning paths (Aggarwal et al., 2021; Jansen and Ustalov, 2019). Moreover, RAG is especially useful for low-resource generation tasks, such as question generation (Yu and Jiang, 2021; Xin et al., 2021; Gu et al., 2019), document-to-slide (Sun et al., 2021), table-to-text (Su et al., 2021), counterargument generation (Jo et al., 2021), entity description generation (Cheng et al., 2020) and text-based games (Murugesan et al., 2021).

Recent research has attempted to reduce hallucinations in LLMs by leveraging external structured knowledge. For example, during fine-tuning, LaMDA (Thoppilan et al., 2022) learns to consult external knowledge sources before responding to the user, including an information retrieval system that can retrieve knowledge triplets and web URLs. Some papers treat the generative model (often large language models) as black-box and retrieve structured information without fine-tuning. For example, BINDER (Cheng et al., 2023) uses in-context learning to output designed API calls that retrieve question-relevant columns from tables.

**Reasoning with knowledge**    By selecting knowledge, reasoning tasks can be solved in a more grounded and interpretable way. To generate an entailment tree explanation for a given hypothesis, Neves Ribeiro et al. (2022) retrieves from textual premises iteratively and combines them with generation. Yang et al. (2022c) proposes a math reasoner that first retrieves highly-correlated algebraic knowledge and then passes them as prompts to improve the semantic representations for the generation task. With the recent advances in LLMs, He et al. (2022a); Li et al. (2023b) retrieve from KG and KB, such as Wikidata, based on reasoning steps obtained from the chain-of-thought (CoT) prompting (Wei et al., 2022).

**Knowledge-grounded dialogue**    Dialogue generation based on relevant tables and knowledge bases has been a practical research application (Wu et al., 2020b; Li et al., 2022b; Nakamura et al., 2022; Gao et al., 2022b; Lu et al., 2023). To tackle the challenge, Li et al. (2022c) and Galetzka et al. (2021) retrieve relevant knowledge, process it into a dense representation and incorporate it into dialogue generation. On top of dense representations, Gu et al. (2020) and Jung et al. (2020) leverage attention mechanisms to flexibly adjust which knowledge to depend on during generation. Some methods (Zhang et al., 2021; Dziri et al., 2021; Chen et al., 2020) first generate subgoals or responses and then use them to retrieve relevant knowledge. The retrieved knowledge then helps amend previous responses. Besides knowledge, Cai et al. (2019b) and Wu et al. (2020a) improve dialogue response generation by retrieving templates or prototype dialogues to augment inputs. Recently, Kang et al. (2023) retrieves relevant subgraphs from KGs, and then utilizes contrastive learning to ensure that the generated texts have high similarity to the subgraphs.

By retrieving from relevant sources, RAG not only improves factuality but also provides the grounding contexts while generating, thus addressing interpretability and robustness concerns. With the potential to handle more information types with recent advances in LLMs (OpenAI, 2023), RAG with structured knowledge could be further enhanced. There are still challenges to be addressed. For example, there could be new designs for better retrieval systems that could promote efficient interactions suitable for diverse knowledge bases. Synthesizing this information correctly is also an open challenge, where it is hard to decide which parts need augmenting in the textual outputs.

### 3.4    Audio

Audio RAG is useful in incorporating audio information in specific audio-language tasks, such as

music captioning, music and text generation, and speech recognition. Moreover, using audio RAG for audio data augmentation has also been proven useful in mitigating the lack of audio-text training data. It could be a promising future direction (Li et al., 2022a).

**Text-audio data augmentation**    For text-audio tasks, one of the most important challenges is the lack of training data on audio-text pairs. Therefore, retrieving audio and textual cues can alleviate the data scarcity problem and improve performance. In audio captioning, which aims at translating the input audio into its description, Koizumi et al. (2020) retrieves guidance captions similar to the input audio from the training set. Then, the retrieved guidance captions are fed into a PLM to help generate new captions, which improves generation performance. To augment scarce speech translation (ST) data, Zhao et al. (2023a) proposes Spoken-Vocab, a technique to convert machine translation (MT) data to synthetic ST data. To form synthetic speech, SpokenVocab retrieves and stitches audio snippets, corresponding to words in an MT sentence. Experiments show that stitched audio snippets can improve translation quality. Kim et al. (2023) leverages a PLM to tackle the data scarcity issue. It retrieves features from the input audio, maps them to continuous vectors using mapping networks, and uses vectors as prefixes for prefix tuning the PLM. With the additional information from retrieved audio, it outperforms previous methods. In text-to-audio generation, Huang et al. (2023a) applies audio-text retrieval to get pseudo text prompts, which enhance audio generation in data-scarce scenarios. To augment the argumentation mining (AM) task in political debates, Mestre et al. (2023) integrates audio features into PLMs, which improves performance when data is scarce.

**Music captioning**    Music captioning is the task of generating a text description or lyrics given the music audio. And RAG is explored to learn better audio-lyric alignment. Manco et al. (2021) proposes the first music audio captioning model, Mus-Caps. Firstly, a pretrained multimodal encoder obtains audio representations that retrieve musical features in the input. As the pretraining bridges the gap between the audio modality and textual understanding, the method improves task performance. He et al. (2022b) learns an audio-lyric alignment through contrastive learning, which results in a higher-quality generation of captions for music.

**Music generation**    Royal et al. (2020) uses deep neural hashing to retrieve music building blocks and then performs generation by using the current music segment to retrieve the next. In automatic speech recognition (ASR), Chan et al. (2023) uses a k-nearest neighbor (KNN) approach to retrieve external knowledge related to the audio and text embeddings. The retrieved knowledge significantly reduces domain adaptation time for ASR.

The audio modality is closely intertwined with other modalities, such as video. Therefore, recent advancements in using audio features for text-video retrieval (Falcon et al., 2022; Mithun et al., 2018) can benefit RAG tasks involving other modalities. Moreover, although audio-text retrieval has been a long-standing task (Liu et al., 2015; Milde et al., 2016a,b), exploring recently discovered techniques (Hu et al., 2022a; Lou et al., 2022; Koepke et al., 2022) could lead to further improvements.

## 3.5    Video

Retrieving video snippets for generation is used primarily in two tasks: video-grounded dialogue and video captioning. Recently, augmenting LLMs with video retrieval also demonstrates good performances, especially in few-shot settings.

**Video-grounded dialogue**    Given video contexts, the model learns to engage in a relevant dialogue. Pasunuru and Bansal (2018) introduces a video-context, many-speaker dialogue dataset, which challenges researchers to develop visually-grounded dialogue models that generate relevant responses from live videos. Similarly, Lei et al. (2020) proposes TVQA+, a dataset that requires retrieving relevant video moments to answer textual questions about videos. Then, it proposes a unified framework that encodes video segments into representations, uses an attention mechanism to locate relevant information, and produces textual answers. To better perform visually-grounded dialogue tasks, Le et al. (2020) retrieves visual cues from prior user queries. The cues are then used as contextual information to construct relevant responses. On video QA, it substantially outperforms prior approaches. Recently, Le et al. (2022) extracts visual cues from the video to augment video-grounded dialogues. The video retrieval is performed with neural module networks, which are instantiated with entities and actions in previous dialogues.

**Video captioning**    Sharing a similar motivation to RAG, Long et al. (2018) first proposes to use attention layers to automatically select the most

salient visual or semantic features and use them to augment caption generation. As a result, it stably outperforms previous methods. (Whitehead et al., 2018) then develops a retrieval-based approach for video description generation. For news videos, it retrieves topically related news documents and then generates a description using a knowledge-aware video description network.

**LLM augmentation** Wang et al. (2022) attempts to augment an LLM to generalize to various video-to-text tasks from a few examples. As the LLMs cannot accept video inputs, it first translates video contents into attributes using image-language models and then prompts the retrieved content to instruct the LLM. It demonstrates good few-shot performances on a wide range of video-language tasks.

Currently, the video-text research bottleneck mainly lies in the representation gap between different modalities. Research has been attempting to learn a better mapping between video-text via joint learning (Xu et al., 2015; Sun et al., 2019). Recent studies on dense video representation learning can also be useful for future video RAG. Besides, some papers (Yang et al., 2023a; Wang et al., 2021a) try to introduce fine-grained interaction between different modalities to learn better aligned representations. Zeng et al. (2022) encourages multiple pretrained models in different modalities to exchange information with each other in a zero-shot manner. Most recently, Zhang et al. (2023a) trains Video-Llama to better align pretrained video and audio encoders with LLM's embedding space.

## 4 Future Directions

With the development of multi-modal LLMs, retrieving multimodal information to augment text generation will be a promising direction to better ground textual generation in real-world contexts, contributing towards building a model that is fully aware and can better interact with the world. Specifically, we describe some potential directions that can be of benefit to the community.

### 4.1 Retrieval Augmented Multimodal Reasoning

One potential application of multimodal RAG is multimodal reasoning. Lu et al. (2022a) first introduces ScienceQA, a large-scale multimodal science question dataset annotated with lectures and explanations. Then, Zhang et al. (2023b) proposes Multimodal Chain-of-Thought (Multimodal-CoT) which incorporates language and vision modalities

into a two-stage (rationale generation and answer inference) framework, surpassing GPT-3.5 by a large margin with a much smaller fine-tuned model. Similar to Zhang et al. (2023b), kosmos-1 (Huang et al., 2023b) breaks down multimodal reasoning into two steps. It first generates intermediate content as the rationale based on visual information and then uses the generated rationale to induce the result. However, both methods may have difficulties understanding certain types of images (e.g., maps), which could be mitigated by retrieving informative image-text pairs.

### 4.2 Building a Multimodal Knowledge Index

In order to facilitate multimodal RAG, one of the most fundamental aspects is building a multimodal knowledge index. The goal is twofold: Firstly, dense representations should support low storage, dynamic updating of the knowledge base, and accurate search. Secondly, it could enable faster search speed with the help of local sensitive hashing (Leskovec et al., 2014), which combats scaling and robustness concerns when the knowledge base is scaled up extremely.

Currently, the dense representations for text snippets are widely studied for documents (Karpukhin et al., 2020b; Gao and Callan, 2021; Gao et al., 2021), entities (Sciavolino et al., 2021; Lee et al., 2021a), and images (Radford et al., 2021a). Besides, there are studies optimizing dense representations in an end-to-end manner (Lewis et al., 2020). Nevertheless, few papers (Chen et al., 2022a) have explored building a multimodal index at the same time for downstream generation tasks. How to map a multimodal knowledge index into a unified space remains a long-term challenge.

### 4.3 Pretraining with Multimodal Retrieval

To better align the abilities to handle different modalities in a pre-trained model, future work could be built on employing retrieval-based approaches during pre-training. Currently, some methods fine-tune the pre-trained generative model to learn to retrieve from different modalities. For example, LaMDA (Thoppilan et al., 2022) calls an external toolset for fine-tuning, including an information retrieval system. Similarly, during fine-tuning, Toolformer (Schick et al., 2023) augments models with API calls to tools including a QA system and a Wikipedia search engine.

When similar retrieval abilities are leveraged during pretraining, the generative models can interact

with retrieval tools much better. Then, instead of relying solely on internal weights, they could effectively use an external base to output more grounded information, provide relevant contexts to users, and update their information accordingly. Such pretraining techniques would also greatly improve robustness for out-of-domain tasks. As an example, Guu et al. (2020a) augments pretraining with an external knowledge retriever, which outperforms previous methods.

To incorporate retrieval with pretraining, there remains the challenge of developing appropriate datasets labeled with retrieval API calls. To tackle this challenge, LaMDA (Thoppilan et al., 2022) uses labels developed by human annotators, which could be expensive to collect. Toolformer (Schick et al., 2023) uses a sampling and filtering approach for automatic labeling, which is inexpensive but could induce bias. A potential solution is to use a neuro-symbolic approach (Davoudi and Komeili, 2021), which uses prototype learning and deep-KNN to find nearest neighbors during training.

## 5 Conclusions

This survey reviews research that augments generative models by retrieving multi-modal information. Specifically, we categorize the current domain into enhancing with different modalities, including image, code, structured knowledge, speech, and video. With the emergence of large multi-modal models, we believe that this survey could serve as a comprehensive overview of an emerging and promising field. Moreover, we hope it could encourage future research in the domain, including retrieval-augmented multimodal reasoning, building a multi-modal knowledge index, and combining retrieval with pretraining.

## 6 Limitations

RAG also has some limitations. For example, there exists an attribution-fluency tradeoff (Aksitov et al., 2023) where the output quality is affected due to the added constraints of the retrieved knowledge.

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

| Modality | ACL | Google | Total analyzed |
|---|---|---|---|
| Image | (67) 17 | 6 | 23 |
| Code | (177) 9 | 24 | 33 |
| Structured | (108) 44 | 11 | 55 |
| Audio | (17) 6 | 14 | 20 |
| Video | (22) 7 | 7 | 14 |
| Total | (291) 83 | 62 | 145 |

Table 1: Paper statistics. Number in parenthesis is the number before manual filtering. "Google" represents searching on google scholar and manually filtering. "Total analyzed" represents the number of total papers after manual filtering

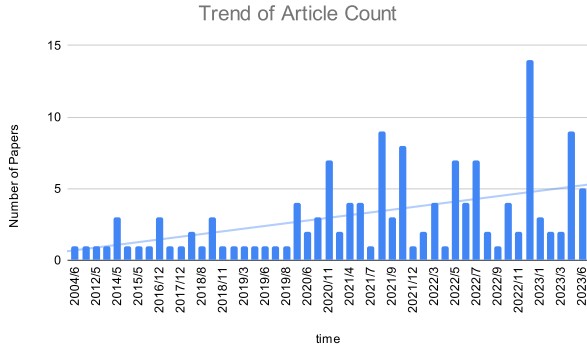

Figure 1: Paper trend analysis

## A Appendix

### A.1 Search Criteria and Results

For searching the ACL anthology articles, we use a keyword search over titles and abstracts. We strictly enforce the keyword "retriev". Then, we enforce either "generat" or "ground" to appear. For each modality, we then add modality-specific keywords: "image" for the image modality, "code" for the code modality, any one from "structured knowledge/table/database/knowledge graph" for the structured knowledge modality, any one from "audio/speech" for the audio modality, and "video" for the video modality.

For searching on Google Scholar, we add the keyword "language models" to select more NLP-related articles. We then perform manual filtering on the top 3 pages of returned results.

The number of retrieved and analyzed research papers can be found in Table 1.

A trend analysis of how the number of papers change across time is shown in Figure 1 We could observe that the domain of multimodal retrieval-augmented generation has indeed developed a lot recently, with peaks reached around end of 2022. The observation is consistent with our hypothesis that multimodal RAG is especially important and helpful in the age of large-scale general-purpose models.