# OpenReview forum: "Retrieving Multimodal Information for Augmented Generation: A Survey"
_EMNLP/2023/Conference — EMNLP 2023 Findings_

### Official Review · Reviewer_yUiZ · 2023-08-05

**Soundness:** 4

**Excitement:**

4: Strong: This paper deepens the understanding of some phenomenon or lowers the barriers to an existing research direction.

**Paper Topic And Main Contributions:**

This survey paper provides a comprehensive review of research on retrieving multimodal information to augment generative language models. The key contributions are:

1. The paper established retrieval-augmented generation with multi-modality as an important emerging group of methods, especially relevant with the rise of large language models.
2. The paper analyzed over 140 recent papers in this domain and drew connections between different works and applications.
3. The paper provided an in-depth literature review of papers using image, code, structured knowledge, audio and video modalities to improve language generation.
4. Finally, The paper discussed future research directions like multimodal reasoning, knowledge indexing, and incorporating retrieval into pretraining.

**Reasons To Accept:**

1. The paper provides a timely, structured, and comprehensive overview of an important emerging research domain: retrieval-augmented generation with multi-modality.
2. The paper analyzed over 140 recent papers in this domain and drew connections between different works and applications.
3. It would be a valuable reference for researchers looking to adapt multimodal retrieval techniques to large language models.


**Reasons To Reject:**

1. It could provide more analysis on the purpose and benefits of different modalities. For example, some (e.g. KGs) are useful to reduce hallucinations, while some (e.g. imagination) are useful to improve generation quality or as a hint.
2. To my understanding, the "multimodal" in the title means the source of retrieval is multimodal, so it would be good to also include the literature of pure text-based RAGs.

**Reproducibility:**

N/A: Doesn't apply, since the paper does not include empirical results.

**Reviewer Confidence:**

3: Pretty sure, but there's a chance I missed something. Although I have a good feel for this area in general, I did not carefully check the paper's details, e.g., the math, experimental design, or novelty.

---

> ### Author Rebuttal · Authors · 2023-08-29
>
> Thank you for the helpful review.
> 1. Currently, we try to illustrate the modality-specific benefits in the beginning of each subsection, such as L183-188 and L438-L442. We will add more analysis on the specific purposes of RAG and link to the nature of each modality in a future version.
> 2. Yes, the term "multimodal" refers to the source of retrieval. Pure text-based RAGs are currently discussed in Section 2.2 (L150-167 in particular) as background knowledge. We can add a more detailed analysis in the revision.

---

### Official Review · Reviewer_ojn8 · 2023-08-06

**Soundness:** 4

**Excitement:**

3: Ambivalent: It has merits (e.g., it reports state-of-the-art results, the idea is nice), but there are key weaknesses (e.g., it describes incremental work), and it can significantly benefit from another round of revision. However, I won't object to accepting it if my co-reviewers champion it.

**Missing References:**

The authors may consider adding related work in the line of text generation:
Zhu, Wanrong, et al. "Visualize Before You Write: Imagination-Guided Open-Ended Text Generation." arXiv preprint arXiv:2210.03765 (2022).

**Paper Topic And Main Contributions:**

This paper reviews methods which assist and augment generative models by retrieving multimodal knowledge, whose formats range from images, codes, tables, graphs, to audio. Their main contribution is a comprehensive survey.

**Reasons To Accept:**

1. Retrieval-augmented generation (RAG) is an important topic.
2. The survey broadly covers mutlimodal RAG papers.

**Reasons To Reject:**

1. There is no major weakness as a survey paper, though more insights and extended future direction section would be appreciated.

**Reproducibility:**

N/A: Doesn't apply, since the paper does not include empirical results.

**Reviewer Confidence:**

3: Pretty sure, but there's a chance I missed something. Although I have a good feel for this area in general, I did not carefully check the paper's details, e.g., the math, experimental design, or novelty.

---

> ### Author Rebuttal · Authors · 2023-08-29
>
> Thank you for the review.
> - We will add more discussion on future directions in the later version.
> - Thank you for the suggestion. The work (Zhu, Wanrong, et al.) uses conditionally generated images to guide generation, instead of retrieval. We will consider adding it to the related work in the revision.

---

### Official Review · Reviewer_4Ruw · 2023-08-11

**Soundness:** 3

**Excitement:**

4: Strong: This paper deepens the understanding of some phenomenon or lowers the barriers to an existing research direction.

**Missing References:**

If the authors focus on broad discussion of *retrieval augmented*, more works in the field of AI creation[1,2], multimodal dialogue[3,4], etc, may be mentioned.
- [1] Neural Storyboard Artist: Visualizing Stories with Coherent Image Sequences
- [2] Multi-Modal Experience Inspired AI Creation
- [3] Maria: A Visual Experience Powered Conversational Agent
- [4] Text is NOT Enough: Integrating Visual Impressions into Open-domain Dialogue Generation
- …

**Paper Topic And Main Contributions:**

This paper reviews recent advances in retrieving multimodal information to augment generation, including using knowledge like image, code, structured knowledge, audio, and video, which provides a unified view for researchers to understand this field.
The main contributions are:
- The authors group retrieval augmented generation with multi-modality and carry out detailed research, analyze the innate connections among these works.
- The author has assessed and discussed the current challenges and proposed some ideas and solutions for potentially promising directions in the future.

**Questions For The Authors:**

See weakness above.

**Reasons To Accept:**

- This work aims to fill the gap by presenting a comprehensive and in-depth research on RAG with multi-modality. It provides a thorough analysis and understanding of RAG with multi-modality, which will be beneficial for the community to gain a deeper insight into its intricacies and details.
- This work is beneficial to both the NLP and multimodal communities, as the questions and insights it presented can stimulate further development in this field.
- This paper has a well-structured organization and good writing skills, making it easy for readers to understand.

**Reasons To Reject:**

- To ensure smooth writing, the authors have included a lot of background knowledge about multimodal learning and other related fields in the paper. This helps readers better understand the topic. However, one issue is that in some places, the introduction of certain works deviates from the main focus of the discussion. For example, from line 663 to line 666, and line 687 to 695, etc. These discussions may dilute the main theme, i.e., RAG with multi-modality and cause confusion regarding some related works.
- One issue is that the definition of multimodality may need to be clarified, as the article considers unconventional structured knowledge such as tables and structured knowledge as multimodal information used for enhancement. This may be hasty, as multimodality typically refers to heterogeneous data. There may be a discussion in the academic community regarding the broad and narrow classifications of multimodality, and this point should be explicitly stated in the paper, at least.
- Further, the distinction of when enhanced information is retrieved needs to be carefully considered. This is because new information from retrieval can occur at various stages, such as data construction, preprocessing, feature extraction, and integration. I don't believe all of these stages can be narrowly categorized as retrieval enhancement. These statements like line 468-475, line 481-489, line 546-547, etc, can be considered to modify.

**Reproducibility:**

N/A: Doesn't apply, since the paper does not include empirical results.

**Reviewer Confidence:**

4: Quite sure. I tried to check the important points carefully. It's unlikely, though conceivable, that I missed something that should affect my ratings.

**Typos Grammar Style And Presentation Improvements:**

The paper is well written with no obvious typos and grammar mistakes. A minor style problem is that the style format of all the grouping name is not consistent. For example, line 230 “Visually grounded dialogue” and line 288 “Text-to-Code Generation”, etc. The capitalization of the first letter should be consistent. The authors are supposed to check all other places to fix this.

---

> ### Author Rebuttal · Authors · 2023-08-29
>
> Thank you for the helpful review. We reply to your concerns here:
>
> - As the video modality has a few RAG studies, we include some papers (e.g. L663-666 and L687-695) that share similar motivations because their tasks and methods could inspire future work in this direction. We now recognize the issue that it may confound our readers and we will adjust these works by clearly pointing out their position and including them as background methods.
> - Thank you for the note. We will put more clarifications on the scope of the survey, especially on structured data. Structured data include knowledge graphs and tables. Graphs are commonly covered in multimodal surveys, such as Guo et al, 2019. In comparison, tables are less seen in multimodal learning, but it is deemed as a heterogeneous source of information by some recent papers, such as UnifiedSKG (Xie et al, 2022) and Uniqorn (Pramanik et al, 2021).
> - Yes, the retrieval can occur at multiple stages. Some retrieve with respect to the inputs, while others retrieve with first-stage outputs. We discussed grouping the methods based on different stages, but a modality-task organization seemed clearer to the readers. We will add more stage-specific analysis for each modality. If needed, we can also add a table that clearly shows the stages in the appendix.
> - Thank you for the additional references, we aim to include all 4 of them in the revision.

---

### Meta-Review · Area_Chair_Vmps · 2023-09-19

**Recommendation:** 4

**Metareview:**

This study investigates the use of multimodality in LLM and how retrieving (inputting) multimodal information contributes to it.

Pros:
It is a useful paper with a good description of the current situation surrounding multimodal LLM, including background knowledge.
A comprehensive study is well done.

Cons:
As reviewer 4Ruw points out, the scope of multimodality covered by this paper needs clarification.

---

### Decision · Program_Chairs · 2023-10-07

**Decision:**

Accept-Findings

**Comment:**

This study investigates the use of multimodality in LLM and how retrieving (inputting) multimodal information contributes to it.

Pros:
It is a useful paper with a good description of the current situation surrounding multimodal LLM, including background knowledge.
A comprehensive study is well done.

Cons:
As reviewer 4Ruw points out, the scope of multimodality covered by this paper needs clarification.